# Dynamic Properties of Di(cyclopentadienecarboxylic Acid) Dimethyl Esters

**DOI:** 10.3390/ijms241914980

**Published:** 2023-10-07

**Authors:** Alojz Anžlovar, Damjan Jan Pavlica, David Pahovnik, Ema Žagar

**Affiliations:** Department of Polymer Chemistry and Technology, National Institute of Chemistry, Hajdrihova 19, SI-1000 Ljubljana, Slovenia; damjan.pavlica@ki.si (D.J.P.); david.pahovnik@ki.si (D.P.); ema.zagar@ki.si (E.Ž.)

**Keywords:** cyclopentadiene, Thiele’s acid ester, regioisomers, NMR spectroscopy, differential scanning calorimetry (DSC), FTIR spectroscopy, GC–MS chromatography, reversible Diels–Alder reaction, dimerization, dynamic chemistry

## Abstract

Di(cyclopentadienecarboxylic acid) dimethyl ester (DCPDME) is a potential dynamic covalent system. When such molecules are used as dynamic crosslinkers in polymers, understanding the reversibility of cyclopentadiene dimerization is crucial to determine optimal melt processing conditions. To this end, we synthesized DCPDME, which consists of three regioisomers with different physicochemical properties, which were investigated by isolating them and further characterizing them using ^1^H NMR, FTIR and DSC. There have been many attempts to improve the synthesis process to increase the reaction yield and purity of isomer 3, and this goal remains a challenge today. In this work, we show that pure isomers 1 and 2 irreversibly convert to the more stable DCPDME isomer 3 at temperatures between 120 and 140 °C in N_2_. This shows that isolation of the pure isomer 3 from the DCPDME isomer mixture is not necessary. The DCPDME isomer 3 is reversibly cleaved to the monomeric cyclopentadienecarboxylic acid methyl ester (CPME), as confirmed with GC–MS and the resulting mass spectrum. The conversion of DCPDME isomers 1 and 2 to isomer 3 was confirmed by heating the synthesized mixture of DCPDME isomers at 135 °C for 5 min in N_2_, producing an almost pure isomer 3 which increased its synthesis yield by 35%.

## 1. Introduction

Cyclopentadiene is a highly reactive diene in Diels–Alder reactions even in the absence of activating electron-withdrawing or electron-donating groups [1]. Since cyclopentadiene itself is relatively unstable, it is always freshly prepared with the thermal decomposition of a more stable dimeric form. Recently, cyclopentadiene and dicyclopentadiene have attracted the attention of the scientific community because of their dynamic properties, although the chemistry of cyclopentadiene is not new. More than a century ago, the German chemist Johannes Thiele [2] performed the reaction of the cyclopentadienyl anion with carbon dioxide, resulting in a mixture of regioisomeric dimers of dicyclopentadiene dicarboxylate [3]. Dimeric acids, which can be isolated by repeated crystallizations, are the result of an endo Diels–Alder coupling between three regioisomers of cyclopentadiene carboxylate [4]. The mixture of these regioisomers is known as Thiele’s acid, which can be converted to the corresponding Thiele’s esters [5]. Cyclopentadienecarboxylic acid methyl esters can exist in three isomeric structures A, B and C (Figure 1I), among which B is the most stable, C is slightly less stable, while A is much less stable [6]. ^1^H NMR shows that only isomers B and C actually exist in the isomer mixture in a mass ratio of 10:1 [4]. Consequently, only isomers B and C are involved in the dimerization reaction. Isomers 1 and 2 are the result of the homodimerization of compounds C and B, respectively, while isomer 3 is formed with the heterodimerization of B and C, with C acting as the diene (Figure 1II). These three reactions actually proceed from 24 possible B–B, C–C, B–C and C–B pairings, as explained by Chen and Wulff [6]. The structure of the dimeric isomer 3 also shows that it is the most stable product from a steric point of view, which is consistent with experimental results [6]. Practically, 60–65 wt. % of the isomer 3 is formed, while the proportions of isomers 1 and 2 are only 12–18 wt. % due to the steric constraints in the dimerization reaction [6].

Immediately after their discovery, there was little interest in these compounds, mainly because of the difficulties in producing a pure isomer 3. This is because the synthetic route involves esterification, subsequent ester hydrolysis and a series of recrystallization steps, all of which result in a low yield [2]. Further studies showed that even after a series of recrystallizations, the final product was still contaminated with other regioisomers [7]. Since Thiele’s acid is poorly soluble in solvents of low and medium polarity, further studies focused mainly on synthetic routes via the corresponding diesters, which are finally converted to the diacid [8,9,10]. There have been many attempts to improve the synthetic process in terms of the reaction yield (up to 50 and 60%) and the purity of isomer 3, but the synthesis of Thiele’s acid or ester is still a challenge today [11]. Valuable information on the development of synthetic approaches and procedures has been reported by Chen et al. [11]. Recently, the diastereomeric salt of Thiele’s acid was prepared with brucine and the absolute stereochemistry of the corresponding diester was determined using X-ray crystallography [10].

Thiele’s acids and the corresponding esters are considered molecular clefts because they are rigid molecules with an inert backbone, while their functionality is projected outward from their central part at defined angles [11]. Potential applications of such structures are in supramolecular systems, conformationally constrained enzyme inhibitors, β-hairpin peptidomimetics and, more recently, in dynamic covalent systems [12,13,14,15]. Dynamic covalent chemistry is a recent descendant of the more familiar supramolecular chemistry, which was developed by transferring dynamic properties from the intermolecular (supramolecular chemistry) to the molecular level [1,2,16,17,18,19,20]. The characteristic of dynamic covalent systems is that they are inherently dynamic, allowing the exchange of molecular components in equilibrium to reach the thermodynamic minimum of the system. Dynamic systems or molecules respond to reaction environments such as physical factors (light, temperature, electrical or mechanical stress, etc.), the reaction medium (solvent) or the presence of stabilization sources (templates, metal ions, protons, etc.). Such dynamic systems are adaptive by nature, which is why dynamic covalent chemistry is also called adaptive chemistry [19].

The widespread potential applications of covalent dynamical systems were derived based on their adaptability. For example, template-driven selection of library members based on host–guest systems induced by supramolecular interactions led to the discovery of new ligands for biomolecules, catalyst systems and molecular receptors [16]. The next potential application based on their adaptability is the development of stimulus-responsive dynamical materials that will play a crucial role in emerging technologies for actuators, sensors and self-healing systems [16]. The latest achievement in dynamic covalent chemistry is the formation of covalently linked complex 2D and 3D molecular architectures through judicious selection of building blocks [19]. The synthesis of adaptable structures and molecules, as well as discrete molecular architectures with novel properties/functions, are promising applications of dynamic covalent chemistry that have yet to reach their full potential. The potential advantage of these covalently bonded structures and molecular architectures compared to supramolecular structures and complexes connected by weak non-covalent (physical) bonds is their higher chemical stability [19].

Many potential dynamic reactions are being studied for the above purposes [16]. One of them is the Diels–Alder reaction. There are several reaction systems based on this type of reaction, such as the maleimide–furan [21], anthracene–olefin [22], anthracene–maleimide [23] and also cyclopentadiene dimerization [24,25]. Compared to the maleimide–furan Diels–Alder reaction, which has been studied extensively, other types have only been studied occasionally. For the cyclopentadiene Diels–Alder reaction, there are few reports of applications in thermally reversible crosslinked polymers or self-healing polymers [26,27,28,29]. Our goal was to study the dimerization of the cyclopentadienecarboxylic acid methyl ester (CPME) and its reversible reaction in order to understand the dynamic properties of di(cyclopentadienecarboxylic acid) dimethyl esters (DCPDME). Understanding the re-versibility of the dimerization of cyclopentadiene is important to determine the optimal melt processing conditions when such molecules are used as dynamic crosslinking sys-tems in polymers.

## 2. Results and Discussion

DCPDME was synthesized according to the procedure described in the literature [11]. The reaction yields a mixture of three isomers (Figure 2), which were separated using column chromatography and designated isomer 1, 2 and 3. The ^1^H NMR spectra of each DCPDME isomer with signal assignments can be found in the Appendix A and agree well with the spectra published in the literature [11,30]. Additionally, ^13^C NMR spectra as well as 2D NMR spectra (COSY, HMBC and HSQC) were recorded and are shown in the Appendix A. In the following section, we studied the thermal properties of the DCPDME isomers with DSC in a nitrogen atmosphere and the changes in the chemical structure of isomers as a function of temperature, using the ^1^H NMR and FTIR spectroscopies, to evaluate the differences in physicochemical properties between three DCPDME isomers.

The DSC curves of the three DCPDME isomers show melting of isomers 1 and 3, while isomer 2 shows no melting process (Figure 1), which is consistent with the appearance of these fractions, as isomer 2 is liquid at room temperature (RT), while the other two are crystalline solids. Furthermore, the DSC curve of isomer 2 was recorded from −80 °C and it shows neither crystallization nor melting below RT. The melting points and melting enthalpies of isomers 1 and 3 differ considerably; isomer 3 has a melting point of 87.7 °C and an enthalpy of melting of 108.6 J/g, while isomer 1 has a melting point of 102.2 °C and a much lower enthalpy of melting (68.3 J/g). All three isomers show additional transitions in the temperature range between 109 and 190 °C (Figure 1). DCPDME isomer 3 shows an endothermic transition with an enthalpy of 29.2 J/g, indicating the decomposition of DCPDME into the monomeric CPME, which is known to be an endothermic process [31]. In contrast, isomers 1 and 2 show exothermic transitions in this temperature range with enthalpies of −53.6 and −37.0 J/g, respectively. Since isomers 1 and 2 also decompose into the monomeric CPME, their DSC curves in this temperature range most likely show the sum of the exothermic (conversion to isomer 3) and endothermic (decomposition) processes. Calculated transition enthalpies in the temperature range from 109 to 190 °C in DSC curves (Figure 1) also contain the enthalpy of vaporization of the CPME, since the boiling temperature of cyclopentadiene carboxylic acid is 212 °C.

In order to identify the origin of the exo-/endothermic transitions of the DCPDME isomers in the range of 110 to 180 °C, all three isomers were heated to 180 °C for 2 min in a DSC calorimeter and the ^1^H NMR spectra of the thermally treated isomers were recorded. The ^1^H NMR spectra of the thermally treated isomers are very similar, indicating that heating to 180 °C leads to the conversion of all three isomers into a compound with approximately the same structure, regardless of which isomer we started from (Appendix A). The ^1^H NMR spectra of the thermally treated isomers are most similar to the spectrum of DCPDME isomer 3, but there are also differences (Figure 2). The three isomers were also thermally treated at 150 °C for a longer time (30 min). The ^1^H NMR spectra of these samples are also comparable; however, the similarity of the spectra to that of isomer 3 is lower. This is most likely a result of the samples being exposed to a high temperature for a longer time, which led to degradation reactions and the formation of various side products such as cyclopentadiene trimer or even higher oligomers [32].

Furthermore, isomers 1 and 2 were thermally treated in a DSC calorimeter at a lower temperature (140 °C) for 5 min in a nitrogen atmosphere and then cooled to RT. Their ^1^H NMR spectra are very similar to the spectrum of isomer 3 (Figure 3), suggesting that when isomers 1 and 2 are optimally thermally treated, they are almost quantitatively converted to isomer 3. The transformation of isomers 1 and 2 into isomer 3 means that isomer 1 or isomer 2 are first cleaved into the monomeric cyclopentadiene methyl esters and subsequently dimerized into isomer 3. It is reported that the cleavage reaction of dicyclopentadiene to cyclopentadiene is ten thousand times slower than dimerization [33,34], and that quantitative cleavage of dicyclopentadiene occurs at 160 °C at normal pressure [35]. Studies of the dissociation of dicyclopentadiene with femtosecond spectroscopy [36] have shown that this process (retro-Diels–Alder reaction) is very fast (reaction time of ~3.0 ps) and that this reaction is proceeded by two reaction mechanisms, i.e., a concerted and a sequential bond-breaking mechanism [37]. In the concerted mechanism, both bonds are cleaved simultaneously, whereas in the sequential pathway, one bond is cleaved first, followed by the cleavage of the second bond. In the dissociation of DCPD, femtosecond spectroscopy has shown that both bond-breaking mechanisms are present in the reaction [37]. These results were obtained with laser pulse-excited/ionized species, while the UV-excited retro-Diels–Alder reaction is ~10 times faster (~200–300 fs) [38].

Finally, a mixture of all three isomers was thermally treated in a DSC calorimeter at 135 °C for 5 min in a nitrogen atmosphere (Figure 4). The comparison of the ^1^H NMR spectra of the mixture before and after thermal treatment and of isomer 3 shows that subjecting it to 135 °C for 5 min are optimal conditions to thermally convert the original mixture of isomers into the almost pure isomer 3. The lower heating temperature (135 °C) is preferred because a smaller amount of side products is formed. These results indicate that DCPDME isomers 1 and 2 decompose during thermal treatment into the monomeric CPME, which reassembles in a reversible reaction to form the more stable DCPDME isomer 3, since only isomer 3 reversibly decomposes into the monomer and back into the dimer (Figure 2). Based on these results, we concluded that, after synthesis, separation of the DCPDME isomers with column chromatography is not necessary, and only other side products formed during the reaction should be removed; only then can the mixture of DCPDME isomers be converted into pure DCPDME isomer 3 using thermal treatment.

The reactions of the dimeric isomers were also studied with FTIR spectroscopy in the temperature range from 60 to 160 °C. The FTIR spectra of the individual DCPDME isomers recorded at 60 °C show minor differences between the isomers due to differences in their structure (Figure 5A). The characteristic absorption bands in the FTIR spectrum of DCPDME isomer 3 are listed in parentheses along with the corresponding bond vibrations. The fingerprint region from 1000 to 1800 cm^−1^ shows absorption bands at 1073 (C–C stretching), 1098 (C–H bending), 1158 (CH_2_ twisting), 1269 (C–C stretching), 1358 (C–H bending) and at 1441 cm^−1^ (CH_2_ scissoring) [39,40]. The absorption bands near 1600 cm^−1^ correspond to the C=C stretching vibration, while the bands between 2800 and 3100 cm^−1^ correspond to the C–H stretching vibration [39,40]. This is the assignment of the main bands in the FTIR spectrum of DCPDME, while the structural differences between the individual DCPDME isomers are too small to be determined with FTIR. Moreover, the strong absorption band between 1710 and 1730 cm^−1^ corresponds to the C=O vibration, since our compounds are di(cyclopentadienecarboxylic acid) dimethyl esters. At 125 °C, the differences between the spectra are smaller (Appendix A), especially in the range from 4000 to 2000 cm^−1^. The similarity between the FTIR spectra increases with increasing temperature (140 °C—Appendix A); at 160 °C, the spectra are almost identical (Figure 5B). A comparison of the FTIR spectra of isomer 3 recorded at 60 °C and 160 °C (Figure 6) shows only minor differences, confirming the earlier conclusions from the ^1^H NMR results, which showed that isomer 3 is not structurally altered upon heating, but only decomposes into the monomeric CPME (Figure 2). The transformation reaction occurs at temperatures between 125 and 140 °C. Therefore, only the DCPDME isomer 3 can be reversibly cleaved into the monomeric CPME and back into the dimer, but no spectroscopic method could prove this. The only evidence that isomer 3 decomposes into the monomeric CPME is the positive transition enthalpy (endothermic reaction) between 109 and 190 °C for this isomer, while isomers 1 and 2 show negative enthalpies (exothermic reaction) in this temperature range. Indeed, the decomposition of DCPDME in the CPME is an endothermic reaction because the dimer is thermodynamically more stable than the monomer. Alternatively, Gas chromatography–mass spectrometry analysis (GC–MS) of DCPDME isomer 3 was carried out using headspace solid-phase microextraction (HS–SPME) at 110 °C. The mass spectrum obtained shows a signal with an *m*/*z* of 124 (Figure 7), corresponding to the molecular ion of the cyclopentadienecarboxylic acid methyl ester. These results confirm that DCPDME isomer 3 decomposes into the CPME at this temperature. In addition, the mass spectrum shows other signals typical of fragments of the CPME, confirming our conclusion. Moreover, the analyzed compound has a retention time of 5.6 min (elution temperature of 96 °C), which also confirms the cleaving of DCPDME to the CPME at 110 °C.

These results show that DCPDME is partially in a dissociated (monomeric) form at a temperature of 110 °C and above. Furthermore, at a temperature of 150 °C, degradation of DCPDME occurs with prolonged exposure, while at 180 °C, some degradation of DCPDME is observed within 5 min. On this basis, we estimate that the appropriate temperature window for processing is between 140 and 150 °C with a short processing time (less than 15 min). Therefore, we conclude that this dynamic DCPDME system is suitable as a crosslinker for linear low-density (LLDPE), low-density (LDPE) and very low-density polyethylene (VLDPE), while it is less suitable for medium- (MDPE) and high-density polyethylene (HDPE) as they require higher processing temperatures.

## 3. Materials and Methods

### 3.1. Materials

Sodium cyclopentadienylide (NaCPD; 2.4 M solution in THF; Sigma-Aldrich, St. Louis, MO, USA,); dimethyl carbonate (DMC; Sigma-Aldrich, St. Louis, MO, USA, anhydrous, >99%); H_2_SO_4_ (Fluka, Buchs, Switzerland, p.a. 95–97%); MgSO_4_ (Sigma-Aldrich, St. Louis, MO, USA, anhydrous, reagent plus, >99.5%); toluene (Sigma-Aldrich, St. Louis, MO, USA, p.a.); isopropanol (Sigma-Aldrich, St. Louis, MO, USA, p.a.); KOH (Kemika, Zagreb, Croatia, p.a., ACS reagent); tetrahydrofuran (THF; Merck, Darmstadt, Germany, p.a. ACS reagent); *n*-hexane (Sigma-Aldrich, St. Louis, MO, USA, p.a., ACS reagent); ethyl acetate (Sigma-Aldrich, St. Louis, MO, USA, p.a., ACS reagent, >99.5%); silica gel 60 (Merck, Darmstadt, Germany, 0.040–0.063 mm, for column chromatography); 4-(dimethyl amino)pyridine (DMAP; Sigma-Aldrich, St. Louis, MO, USA, ReagentPlus, >99%); HCl, (Fluka, Buchs, Switzerland, p.a., ACS reagent, fuming, >37%, APHA < 10).

### 3.2. Synthesis of Di(cyclopentadienecarboxylic Acid) Dimethyl Esters (DCPDME)

The synthesis was performed according to the synthesis procedure described in the literature, with some modifications [11]. A sodium cyclopentadienylide (NaCPD) solution in THF was weighed into a flame-dried (10 min with burner in the flow of Ar) round-bottomed flask (equipped with a magnet) in a dry box and sealed with rubber septa. In a separate Erlenmeyer flask, an equivalent amount of dimethyl carbonate (DMC) was weighed in and sealed with rubber septa in a dry box. After both flasks were removed from the dry box, the THF (11.1 mL) was added to DMC with a syringe and the solution of DMC in THF was then added to NaCPD in the round bottom flask with a syringe. This mixture was then placed on a magnetic stirrer and heated to 60 °C for 24 h while stirring. The next day, the mixture was cooled and concentrated in vacuum. The resulting solid was washed with diethyl ether with vacuum filtration and dried overnight in a vacuum. The result was an air-sensitive, light brown solid. The resulting product was quickly transferred to a flame-dried round-bottomed flask equipped with a magnet, and was then purged with Ar and sealed with rubber septa. A solution of H_2_SO_4_ in isopropanol was then added to this flask with a syringe. When the solid was suspended in the medium, the mixture was heated on a magnetic stirrer at 50 °C for 18 h. After cooling, the isopropanol was removed on a rotary evaporator and the resulting viscous liquid was dissolved in 100 mL of toluene. This solution was extracted three times with 100 mL of deionized H_2_O, dried with MgSO_4_ and was finally filtered to remove the desiccant. The resulting solution was concentrated on a rotary evaporator and placed on a silica gel column. Column chromatography was performed using the hexane/ethyl acetate solvent system of 20/1 to give three fractions corresponding to the three isomers of DCPDME. The solvent was removed with a rotary evaporator, and the final products (three isomers) were dried overnight in vacuum at 45 °C.

### 3.3. Characterization Methods

Differential scanning calorimetry (DSC) curves were measured on a DSC-1 calorimeter (Mettler Toledo, Greifensee, Switzerland) in the temperature range from 25 °C to 190 °C with a heating rate of 10 K/min in N_2_ atmosphere.

The changes in the chemical structure of DCPDME were studied by Fourier transform infrared spectroscopy (FTIR) on a Spectrum One FTIR spectrometer (Perkin-Elmer, Waltham, MA, USA) in transmission mode in the spectral range between 400 and 4000 cm^−1^ and with a spectral resolution of 4 cm^−1^. The samples were prepared using the KBr pellet technique. The SPECAC temperature cell was used for studies in the temperature range from 25 to 160 °C.

Proton nuclear magnetic resonance (^1^H NMR) spectra of synthesized DCPDME were recorded using a Bruker 600 MHz spectrometer (Billerica, MA, USA) under the following quantitative conditions: pulse at 90°, pre-scan delay time of 6.5 μs, acquisition time of 2.75 s or 2.0 s. Samples were dissolved in DMSO-*d*_6_ or in CDCl_3_. Initially, the ^1^H NMR spectra of DCPDME isomers were recorded in CDCl_3_ for an easier comparison with the spectra in the literature [11].

Gas chromatography–mass spectrometry analyses (GC–MS) were performed using a GC 6890N gas chromatograph (Agilent, Santa Clara, CA, USA) coupled to a 5973 Network mass spectrometer (Agilent, Santa Clara, CA, USA). Headspace solid-phase microextraction (HS–SPME) at 110 °C was used for sample injection. Helium was used as the carrier gas. The oven temperature was initially held at 70 °C for 2 min, then increased to 280 °C at a heating rate of 20 °C/min and then held at this temperature for 0.5 min. A detailed description of the complete GC–MS analysis is given in the Appendix A.

## 4. Conclusions

Di(cyclopentadienecarboxylic acid) dimethyl esters (DCPDMEs) were synthesized and three regioisomers were isolated with column chromatography and identified with ^1^H NMR. By physical appearance, isomers 1 and 3 are crystalline solids, while isomer 2 is a liquid at room temperature. DCPDME isomer 1 melts at 102.2 °C with a melting enthalpy of 68.3 J/g, while isomer 3 melts at 87.7 °C with a melting enthalpy of 108.6 J/g. All three isomers show additional processes in the temperature range from 109 to 190 °C. In the case of isomers 1 and 2, these transitions are exothermic with enthalpies of −53.6 and −37.0 J/g, respectively, while isomer 3 shows an endothermic transition with an enthalpy of 29.2 J/g.

^1^H NMR spectroscopy of the thermally treated DCPDME isomers 1, 2 and 3 at 140 °C for 5 min in an N_2_ atmosphere showed that isomers 1 and 2 were irreversibly converted to the more stable isomer 3 of a dimeric DCPDME upon heating. FTIR experiments using a thermal cell confirmed the results of the NMR experiments, which is that isomers 1 and 2 were converted to isomer 3 when heated to 140 °C for 5 min.

Based on the DSC, NMR and FTIR experiments, we therefore conclude that the three isomers of the dimeric cyclopentadienecarboxylic acid methyl ester have different physicochemical properties. Isomers 1 and 2 are irreversibly converted upon heating to the more stable isomer 3 of the DCPDME at temperatures between 120 and 140 °C, which causes the exothermic transitions in their DSC thermograms. Only DCPDME isomer 3 can be reversibly cleaved into the monomeric cyclopentadienecarboxylic acid methyl ester (CPME) at temperatures between 109 and 190 °C, which explains the endothermal reaction enthalpy of isomer 3, as DCPDME is more stable than the monomeric CPME. In addition, a GC–MS analysis was performed using an SPME headspace injection system at 110 °C. The obtained mass spectrum showed the ion peak of the cyclopentadienecarboxylic acid methyl ester at *m*/*z* = 124, thus confirming the decomposition of DCPDME into the CPME at this temperature.

Finally, a synthesized mixture of DCPDME isomers was thermally treated in DSC at 135 °C for 5 min in a nitrogen atmosphere. The ^1^H NMR spectrum of the thermally treated isomer mixture confirmed the conversion to the almost pure DCPDME isomer 3, which increased the synthesis yield of this isomer by 35%. Based on these results, we estimate that the suitable temperature window for melt processing of the PE and DCPDME mixture is between 140 and 150 °C with a processing time of a few minutes. At higher temperatures, there is a risk of DCPDME degradation. Our conclusion is that the dynamic DCPDME crosslinker is suitable for LLDPE, LDPE and VLDPE, while it is less suitable for MDPE and HDPE.

## Data Availability

Data is contained within the article or Appendix A.

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
