# Peer review of "Dynamic Properties of Di(cyclopentadienecarboxylic Acid) Dimethyl Esters"

_ijms, 2023, doi:10.3390/ijms241914980_

Round 1

Reviewer 1 Report

The manuscript is about the synthesis and purification of 3 isomers of Dicyclopentadiene dimethyl ester (DCPDME) as a potential diene to be used in several different cycloaddition reactions. The isomers were purified and characterized deeply and their stability were investigated upon heat treatment. Based on the literature, the need for pure form of Isomer 3 seems to be addressed with this study, showing the transition of isomer 1 and 2 to 3 upon heating. For the representaion of the results, following points could be addressed so it would be more understandable 

NMR peaks can be enlarged and similar/different peaks or the peaks pointing the transition could be highlighted on the spectra 

FT-IR pekas could be assigned, at least for the major characteristic peaks

The manuscript is about the synthesis and purification of 3 isomers of Dicyclopentadiene dimethyl ester (DCPDME) as a potential diene to be used in several different cycloaddition reactions. The isomers were purified and characterized deeply and their stability were investigated upon heat treatment. Based on the literature, the need for pure form of Isomer 3 seems to be addressed with this study, showing the transition of isomer 1 and 2 to 3 upon heating. For the representaion of the results, following points could be addressed so it would be more understandable 

NMR peaks can be enlarged and similar/different peaks or the peaks pointing the transition could be highlighted on the spectra 

FT-IR pekas could be assigned, at least for the major characteristic peaks

Author Response

Journal: International Journal of Molecular Sciences

Manuscript ID: IJMS-2607118

Manuscript Title: Dynamic Properties of Dicyclopentadiene Dimethyl Esters

Authors: Alojz Anžlovar, Damjan Jan Pavlica, David Pahovnik and Ema Žagar

Response to comments of reviewer 1

The manuscript is about the synthesis and purification of 3 isomers of Dicyclopentadiene dimethyl ester (DCPDME) as a potential diene to be used in several different cycloaddition reactions. The isomers were purified and characterized deeply and their stability were investigated upon heat treatment. Based on the literature, the need for pure form of Isomer 3 seems to be addressed with this study, showing the transition of isomer 1 and 2 to 3 upon heating. For the representaion of the results, following points could be addressed so it would be more understandable: 

NMR peaks can be enlarged and similar/different peaks or the peaks pointing the transition could be highlighted on the spectra. 

FT-IR pekas could be assigned, at least for the major characteristic peaks.

Answer: The 1H NMR spectra in Figures 2, 3, and 4 were magnified to make low intensity signals visible. In Figure 4, the 1H NMR signals of isomers 1 and 2 in the DCPDME isomer mixture were marked with the arrows. Additionally, the 1H NMR spectra in figures S1 and S17 in the Supplementary Materials were also enlarged to expand the peaks of lower intensity.

The main characteristic absorption bands of DCPDME in the FT-IR spectra (Figure 5 and Figures S6 and S7) have already been assigned in the original manuscript (page 8, paragraph 1). The absorption bands due to structural differences between the three isomers of DCPDME can not be assigned unambiguously because the structural differences between the isomers are rather small. A sentence on this topic was added to the revised manuscript.

Reviewer 2 Report

The review report is attached below.

Author Response

Journal: International Journal of Molecular Sciences

Manuscript ID: IJMS-2607118

Manuscript Title: Dynamic Properties of Dicyclopentadiene Dimethyl Esters

Authors: Alojz Anžlovar, Damjan Jan Pavlica, David Pahovnik and Ema Žagar

Response to comments of reviewer 2

In this manuscript, the authors investigated the behavior of DCPDME regioisomers, particularly their dimerization and the reversibility of this process. They also mentioned the challenge of synthesizing isomer 3 with higher yield and purity. This study provided experimental data on the conversion of isomers 1 and 2 to the more stable isomer 3 at temperatures between 120 and 140 °C in a nitrogen atmosphere and the reversibility of isomer 3 to the monomeric cyclopentadiene methyl ester. Overall, this research indicates that the dynamic behavior of DCPDME isomers could be harnessed to simplify the synthesis process and potentially increase the yield of the desired isomer 3. However, I do have major concerns regarding this manuscript and regretfully could not recommend its publication in IJMS.

  1. Lack of novelty and originality: The synthesis and characterization of DCPDME was well studied by Chen and coworkers in J. Org. Chem. 2015, 80, 8979−8989. The reversibility of dimerization of cyclopentadiene has also already been studied extensively by researchers in the field. Please refer to the recent review published in Chem. Rev. 2021, 121, 6777−6801.

Answer: We agree with the reviewer that the synthesis of DCPDME has been well studied by Chen et al. and  many other groups, however the dynamic properties and reversibility of DCPDME isomers have not been extensively studied by researchers in the field, as can be seen from the review paper by Levandowski and Raines, which contains only a short paragraph on dimeric cyclopentadiene esters with only basic data on the synthesis, while the dynamic properties are only briefly mentioned. Therefore, the reversibility of DCPDME isomers has not yet been studied or at least it is not reported. We have not found any reports of DSC studies on these compounds in the literature. It is also not reported that isomers 1 and 2 of DCPDME can be readily converted to isomer 3 by a simple thermal treatment in a nitrogen atmosphere, which is important for the future synthesis and application of these compounds. From this point of view, our manuscript is novel and original.

  1. As discuss in the manuscript, in figure 3, the authors heated DCPDME isomers 1 and 2 to 140 °C and observed that these isomers converted to isomer 3 through the changes in NMR spectra. It is suggested that the authors also need to present the NMR spectra of isomer 1 and 2 before heating in DMSO-d6 despite the authors already provided NMR spectra for isomers 1–3 in CDCl3. Using DMSO-d6 will lead to different splitting and chemical shifts compared to CDCl3 so the readers can see the changes more clearly in NMR spectra using the same solvent. Moreover, 2D NMR spectra for each isolated compound (1 H, 13C, COSY, NOESY, HSQC, HMBC, TOCSY) should also be provided for full characterization purpose.

Answer: Figures S2, S3 and S4 in the Supplementary Materials have been replaced with Figures representing 1H NMR spectra of DCPDME isomers recorded in DMSO-d6. In addition, we have included in the revised manuscript 13C NMR spectra and COSY, HMBC, HSQC 2D NMR spectra of DCPDME isomers (Figures S5 – S16 in the Supplementary Materials).

  1. The biggest impact this study could bring to the field is its application in dynamic covalent systems. The authors should conduct experiments to demonstrate how this finding could have implications in optimizing melt processing conditions when using DCPDME as dynamic crosslinkers in polymers. However, this very important part is missing in this manuscript.

Answer: Unfortunately, such experiments would take some time to carry out and they require a lager quantity of dynamic cross-linker, which can not be done in such a short time intended for the correction of the manuscript. The processing temperature of polyethylene (PE) is between 140 and 180 °C and is related to the melting temperature of PE, which is a function of its density and molar mass. Our results show that at temperature of 110 °C and above, DCPDME is partiallly in dissociated (monomeric) form and that DCPDME degrades above 150 °C, implying that the appropriate processing temperature is between 140 and 150 °C. Therefore, we conclude that this dynamic crosslinker is suitable for medium density, low density, and very low density polyethlene, while it is less suitable for medium and high density polyethylene. This text has also been included in the manuscript at the end of Results and Discussion and at the end of Conclusion sections.

Round 2

Reviewer 2 Report

Happy with the author's revision. Thank you for your time and effort.